# LabNet hardware control software for the Raspberry Pi

**Alexej Schatz\*, York Winter**

Humboldt Universität, Berlin, Germany

**Abstract** Single-board computers such as the Raspberry Pi make it easy to control hardware setups for laboratory experiments. GPIOs and expansion boards (HATs) give access to a whole range of sensor and control hardware. However, controlling such hardware can be challenging, when many experimental setups run in parallel and the time component is critical. LabNet is a C++ optimized control layer software to give access to the Raspberry Pi connected hardware over a simple network protocol. LabNet was developed to be suitable for time-critical operations, and to be simple to expand. It leverages the actor model to simplify multithreading programming and to increase modularity. The message protocol is implemented in Protobuf and offers performance, small message size, and supports a large number of programming languages on the client side. It shows good performance compared to locally executed tools like Bpod, pyControl, or Autopilot and reaches sub-millisecond range in network communication latencies. LabNet can monitor and react simultaneously to up to 14 pairs of digital inputs, without increasing latencies. LabNet itself does not provide support for the design of experimental tasks. This is left to the client. LabNet can be used for general automation in experimental laboratories with its control PC located at some distance. LabNet is open source and under continuing development.

## Editor's evaluation

LabNet is an exciting new platform for experimental control using Raspberry Pis. As experiments get more complex in neuroscience, new validated tools are needed to continue to allow users flexibility, precision, and be fast, and LabNet is such a tool. Through extensive benchmarking and documentation of their tool, they demonstrate excellent performance, scalability, and provide examples of how their platform can be adopted.

**\*For correspondence:**
alexej.schatz@hu-berlin.de

**Competing interest:** The authors declare that no competing interests exist.

## Introduction

The combination of open-source software, low cost microcontroller electronics, and the easy access to digital fabrication have led to a plethora of open-source solutions for animal behaviour experimental systems (Open Behaviour [*Laubach et al., 2021*], Bpod [*Sanders, 2021*], Autopilot [*Saunders and Wehr, 2019*], pyControl [*Akam et al., 2022*], MiniScope [*Cai et al., 2016*], Bonsai [*Lopes et al., 2015*], Whisker [*Cardinal and Aitken, 2010*], OpenEphys GUI [*Siegle et al., 2017*]). Using our 10-year experience with the Rasperry Pi for animal behaviour experimental control and after two decades with different self-developed embedded control approaches, we have developed in C++ a new, powerful, and highly versatile platform for hardware control via the Raspberry Pi.

We had two major goals: the platform has to be suitable for time-critical operations and be easy to extend. Furthermore, LabNet had to support a wide variety of hardware components and we wanted to simultaneously control multiple animal behaviour experimental operant boxes. When conducting automated behavioural experiments, it is advantageous to test many animals in parallel with identical or, if necessary, individually specific experiments. This is the only way to obtain complete data sets quickly and can only be achieved through automation. *Figure 1* shows examples of operant

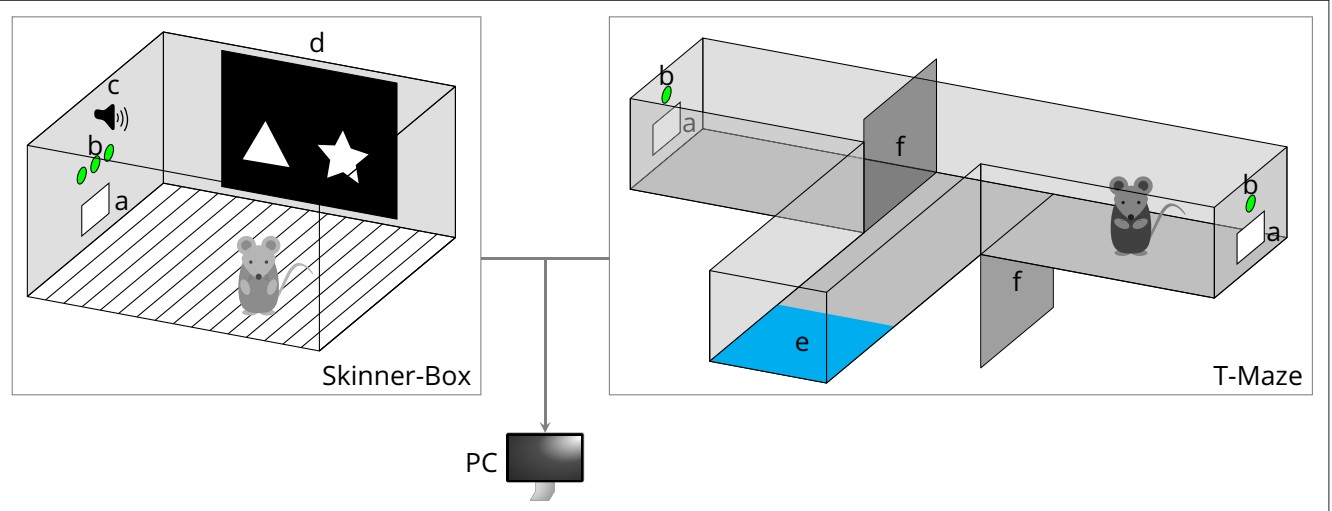

**Figure 1.** Examples of behavioural setups controlled by LabNet. A Skinner box (left) contains (**a**) a feeder magazine that typically has a photo gate for nose-poke detection and a reward pellet dispenser. It also has (**b**) a row of LEDs and (**c**) a tone generator. (**d**) A monitor displays visual stimuli and may have a touch sensor for touchscreen functionality. The T-Maze (right) also has (a) a food magazine and (b) LEDs, and furthermore (**e**) an optical sensor to detect the return of the mouse to the start position and (**f**) two motorized doors that can be lowered to restrict access to the arms. Legend: Images and diagrams generated with TikZ.

conditioning cages (Skinner boxes; *Skinner, 1938*) as controlled by LabNet. Our intention was not to create a completely new ecosystem like Autopilot. We wanted to simplify communication with hardware for projects using their coding language of choice on the PC. Also, we wanted to remain general so that LabNet can become a general platform for experimental laboratory automation.

We selected the Raspberry Pi because it is low cost, powerful, has a wide selection of I/O add-ons and software components. To keep signal lines short, we gave each experimental setup its own Raspberry Pi. All systems are connected to the Ethernet network and are manageable via a central instance. This instance can be a normal PC and be located outside of the laboratory. Thus, the condition of experimental states and the animals can be monitored at any time, even without entering the laboratory.

Autopilot also uses a swarm of Raspberry Pis. However, it implements a hierarchy where each Raspberry Pi can take a different role. This requires an additional configuration step and can complicate troubleshooting. We wanted to avoid this as well. This is the reason why each of our Raspberry Pis runs the same software and overall experimental control is executed by the central instance that is for example run from a PC. This separation also determined the network architecture of the entire system. The local instances on the Raspberry Pi are servers, and the central instance is the client. There can be one client to control experiments on all systems or multiple clients each controlling an experiment on one or more systems. But, at least so far, not multiple clients connected to one server.

## Results

### System overview

We designed LabNet as a distributed system (network) where LabNet presents a node running on a RasPi. We had two important requirements for this system: openness and scalability (see *van Steen and Tanenbaum, 2017*). Openness means that each node can control an experimental chamber on its own or together with a number of other nodes (for experimental system examples, see *Figure 1* and 4). Scalability means that there can be any number of nodes and thus experimental chambers in the system. However, a node or a chamber has to be removable from the system without adjustments on the other nodes. To ensure this, each node in our system is controlled by a RasPi, each RasPi is configured in the same way and controlled by the same software. However, this also comes with the restriction that at most one experimental system can be connected to each RasPi to be removable without electrical adjustments. But this also means a simplification: LabNet only needs to accept a

single connection and does not need resource management for multiple connections, because only one experiment runs on one system and the hardware is not shared.

Thus, the network of LabNet nodes represents the distributed system and offers, as servers, the hardware resources in the network. However, hardware control in the context of the experiments is the responsibility of clients and not a LabNet duty. For example, LabNet does not decide about an output pin state, but LabNet knows how to switch the state and performs it at the client's request. One client could take the control over of the entire LabNet distributed system or divide the nodes among several clients. It all depends on the situation and requirements: a large number of identical experimental chambers with identical experimental tasks are usually controlled by one client while different experimental tasks may better be controlled by separate clients, also to start and stop experiments independently. For communication between LabNet and client a flexible and fast message protocol using Protobuf was developed (section Message protocol). The clients can be implemented in any language with Protobuf support, for example, Python, C#, C++, etc.

Since the Raspberry Pi is a single-board computer, it runs 'Raspberry Pi OS': a Debian-based Linux distribution. This allows a large freedom in the choice of programming language and software tools. Both interpreted languages, such as Python, and compiled languages are available. LabNet was required to meet two criteria:

1. Time-critical: all operations should be performed as quickly as possible.
2. Flexible: new functionality extensions should be as simple as possible.

Unfortunately, all interpreted programming languages have a disadvantage in execution speed compared to compiled languages. Nevertheless, many of the tools developed recently, such as Autopilot and pyControl, use Python. Python is a simple language and provides many packages for all purposes. However, because execution speed was of primary importance we decided to use C++.

Extensions with new functionalities is generally possible in two ways: (i) software adaption with recompilation in case of compiled languages and (ii) a plug-in system. In the current LabNet version, we use recompilation but our road map also includes a future plug-in system. To simply modifications the software must have a suitable architecture and a high degree of modularization.

Since its version 2 the Raspberry Pi has 4 cores. In addition, most of its hardware controllers, such as USB or Ethernet, operate asynchronously, thus they do not require CPU capacity because of DMA (Direct Memory Access), and they report their work completion via interrupt messages. LabNet needed an architecture that optimally leverages this already available hardware asynchrony for parallel execution. Handling GPIO lines is fast, but accessing a UART may lead to considerable delays in sequentially executed software. This presupposes the use of multiple threads. Since programming with many parallel threads is a very error-prone and time-consuming task, we decided to develop an actor-based software (see sections Actor model and SObjectizer). This also provides higher flexibility and software modularity.

## Example

The following example and the corresponding listings (1–3) show how a client can initialize and control the hardware together with a LabNet server on a RasPi with a simple hardware setup. The client could run on a PC and use any language that has support for Protobuf-like Python, C++, C#, etc. Since we use C# in our experiments, the C# notation is also used in the listings. Basically, it shows the use of some of the LabNet messages, but the communication via TCP/IP is omitted for simplicity. We simply assume that a TCP/IP client exists and handles all operations like send, receive, and serialization.

Let us assume an experimental setup with an LED and audio as stimuli, a valve to release a liquid reward and a photo gate as a nose-poke sensor to detect animal behaviour. All these components can be connected directly to the GPIO pins via a simple circuit. The headphone jack can be used for audio output. Then, we need to send five commands to LabNet to initialize all components; see Listing 1. It would usually be necessary for the client to wait for the responses from LabNet and check the initialization results. Here, we skip this step.

During experiments, animals must usually perform some operant behaviour. This can be as simple as nose poking to trigger a photo gate after a certain stimulus has been perceived. In Listing 2, LabNet activates an LED and produces a sine tone. In the case of the tone, it is instructed to automatically generate a pulsed output. On detecting On and Off state changes,

LabNet transmits such photo gate state changes to the client. In response to the photo gate state change, a reward can be provided. In Listing 3, a liquid reward valve is opened for 100 ms.

A typical experiment in combination with LabNet comprises several phases:

- establishing a TCP/IP connection;
- initializing all hardware components;
- turning stimuli on or off in a specific order;
- waiting for an animal reaction and potentially providing a reward.

## Performance evaluation

Because the neurons in the brain work in the millisecond range, the response times in behavioural experiments are critical and should match that range.

```
// start GPIO interface with WiringPi
var initIo = new GpioWiringPiInit();
// init a digital output on pin 5
var led = new GpioWiringPiInitDigitalOut {
    Pin = 5,
    IsInverted = false
};
// init a digital output on pin 26
var valve = new GpioWiringPiInitDigitalOut {
    Pin = 26,
    IsInverted = false
};
// init a digital input on pin 23
var poke = new GpioWiringPiInitDigitalIn {
    Pin = 23,
    IsInverted = false,
    ResistorState = PullUp
};

// start sound interface
var initSound = new InitSound();
// create a sine tone
var sine = new DefineSineTone {
    Id = 1,
    Frequenz = 1000,
    Volume = 0.5
};
```

## Listing 1
Each generated object represents an initialization message to be serialized with Protobuf and transmitted to LabNet. The first message initializes the digital I/O interface with WiringPi pin notation. The next three initialize an LED, a valve, and a poke sensor on the WiringPi interface. The fifth creates the sound generator on the headphone jack. The last, initializes a sine tone with 1 kH frequency and 50% volume. Object initialization in C# notation. Serialization and TCP/IP communication not listed.

```
// change the state of the pin 5 to true
var setLed = new DigitalOutSet {
    Id = new PinId { Interface = GpioWiringpi, Pin = 5 },
    State = true
};
```

```
// turn the sine tone in pulses of 500ms on and off
var pulseSound = new DigitalOutPulse {
    Id = new PinId { Interface = Sound, Pin = 1 },
    HighDuration = 500, // ms
    LowDuration = 500, // ms
    Pulses =10
};
```

### Listing 2

Information for transmission via Protobuf, building on the initialization from Listing 2.2. The LED is set to ON state (until an OFF command). A sound, defined as 1 kHz in Listing 2.2, is emitted as 10 pulses of 500 ms. Object initialization code in C# notation.

```
DigitalInState pokeState; // new poke state from LabNet
if (pokeState.State) // if new state true -> on
   // give a reward
   var reward = new DigitalOutPulse {
       Id = new PinId { Interface = GpioWiringpi, Pin = 26 },
       HighDuration = 100, // ms
       Pulses = 1
   };
}
```

### Listing 3

In this example, LabNet has transmitted via Protobfuf a new poke sensor state (pokeState) to the PC. If the new poke state is true, a new message directs LabNet to deliver a reward by opening valve at pin 26 for 100 ms, as initialized in Listing 1. Code shown in C# notation.

LabNet was subjected to three tests to determine the latency times when executing different commands. A RasberryPi was connected to a PC via a router. For all tests, the client ran on a Linux PC (Ubuntu 20.04, Intel Core i7-6700 3.4 GHz with 16 GB RAM). To allow a comparison, the client was implemented in three languages: Python, C#, and C++. We used Python version 3.8. Python tests ran directly on top of the socket, synchronously with no additional software layers. In Python, it is also essential to deactivate Nagle's algorithm. The C# version was implemented under .NET 6 and used Akka.NET, an actor framework. For C++, we used GCC 9.4.0, Boost version 1.75, and SObjectizer 5.7.2. Thus, all tests in C# and C++ were implemented as actors inside an actor framework. The source code of all tests is included in the GitHub repository under 'examples'.

As always, all benchmarks must be interpreted with a certain degree of caution. For example, it is generally not possible to create the same initial situation for the implementations in all languages. With C++, an external library like Boost must be used for communication via TCP/IP. In addition, we used SObjectizer for C++ and Akka.NET for C# for asynchronous message processing. This theoretically gives the implementation in Python a slight advantage, as it runs synchronously and also has no complex calculations. Performance problems usually occur in Python code whenever true parallel execution is required (because of the Global Interpreter Lock) or when the calculations cannot be outsourced to a library implemented in C. Nevertheless, all three implementations provide a reasonable expectation of the latency in real cases. This is especially true for C# and C++, since they run asynchronously and thus simulate the execution of several parallel-running experimental tasks with animals in a first approximation.

LabNet was also compiled with different optimizations to investigate the performance effects of the different RasberryPi boards. However, GCC 8.3 was used for all versions. The first version had only the default release optimizations from CMake and could run on all RasPi boards. The optimization flags were as follows:

- `-mcpu=cortex-a7` for Pi 2;
- `-mcpu=cortex-a53` for Pi 3;
- `-mcpu=cortex-a72` for Pi 4;

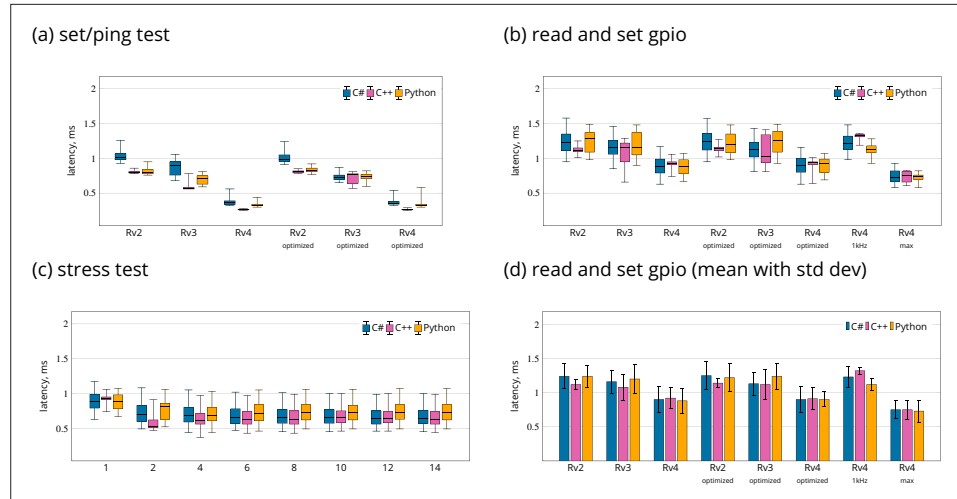

**Figure 2.** Results from LabNet performance tests. (**a**) Time to set a digital output as a Ping equivalent. (**b, d**) Latency to set a digital output in response to a change on a digital input. (**c**) Run the 'read and set' test for up to 14 IO-pin pairs in parallel. LabNet ran on RasPi 4. Tests were repeated 10,000 times and results are in milliseconds. Tests were performed on three different RaspberryPi boards: Rv2 is 2B, Rv3 is 3B+, and Rv4 is 4B with 1 GB RAM. *optimized* refers to the LabNet with some additional optimization flags (see main text). 1 kHz refers to the version without optimizations running on Rv4 with 1 kHz polling and *max* refers to non-stop polling. Box plots in (a–c) show median, lower and upper quartile, and whiskers the 2.5th and 97.5th percentiles. Data in (**d**) given as means and STD.

- `-mfpu=neon-vfpv4  -mfloat-abi=hard` - floating-point number optimizations for all versions.

Each test was run 10,000 times and was performed on three different RaspberryPi boards: RasPi 2B, 3B+, and 4B with 1 GB RAM. Statistical variables such as: mean, STD, median, and percentiles were calculated from the time measurements.

## Set digital out test

In the 'set out test', a digital output is alternately set to 0 and 1. After the command for setting the pin has been received and processed, LabNet automatically sends back an acknowledgement. In this test, the time between sending the set command and receiving this confirmation was measured. Because of the simplicity, this test can also be seen as a type of ping measurement.

The set command from the client has 10 bytes. The server response is 22 bytes long, and includes the execution timestamp.

The results are shown in *Figure 2a*. The median for C# was between 0.36 ms for 4 and 1.01 ms for 2. For Python 0.32 on RasPi 4 and 0.80 on RasPi 2. For C++ 0.26 on RasPi 4 and 0.80 on RasPi 2.

## Read and set GPIO

In this test, the reaction time to external events was measured. LabNet first had to detect the interruption of a photogate by an animal's nose, send a message to the client and in response the client had to initiate the change of a digital output state through a message to LabNet. To simulate the nose-poke events, a second RasPi was used. Two pins between the two RasPis were connected: one for the test signal from the second RasPi and one for the response from LabNet. The second RasPi was only responsible for switching the first pin to 1, stopping the time and waiting until the RasPi with LabNet had also switched the second pin to 1 in response. The time between these two high events is the latency (see *Figure 3b*). The measurement software on the second RasPi was written in C++ and ran on RasPi 3B+. We also verified how fast this software can detect the response signal. To do this, we simply connected test and response pins on the second RasPi together. This way the response pin goes immediately high if the test pin is set. The latency in this case was only 0.7 ± 0.6 µs. This means that the second RasPi acts like a 1 MHz oscilloscope. This is entirely sufficient in our case.

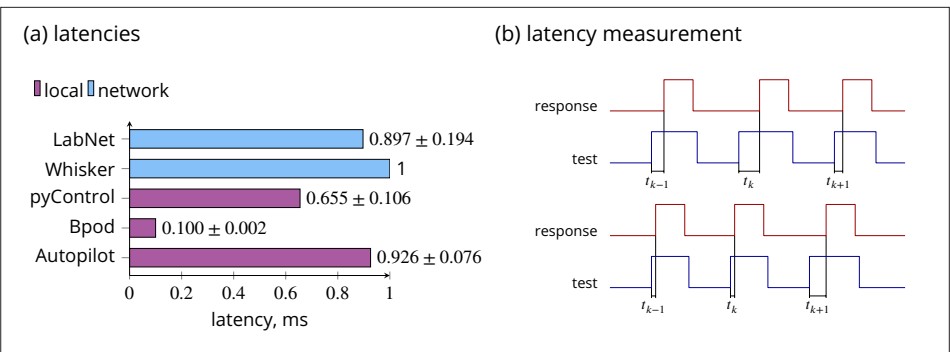

**Figure 3.** Latencies comparison and measurement. (**a**) Comparison of execution latencies. All tools performed the same 'read and set' task to achieve comparability, except Whisker. Whisker server implements a 1 kHz polling frequency on the PC. LabNet for digital input polling depends on the internal RasPi 4 kHz polling frequency. Only for LabNet do the latencies include the message transfer over the Ethernet wire. Values give means with STD. LabNet was operated with a C# client. LabNet and Autopilot use the RasPi 4. (**b**) The latency measurement in Read and set GPIO test. The measurement RasPi generates the high 'test' signal, saves the time, and waits until the 'response' signal is also high. The time $t_k$ between these two high events is the latency. The test RasPi repeats this for 10,000 times and saves the results in a CSV file. RasPi acts here as 1 MHz oscilloscope. All packages (Autopilot, Bpod, pyControl, and LabNet) were tested in the same way. In the Stress test we have multiple 'test' and 'response' lines.

The digital input state message from LabNet is 22 bytes long, and includes the timestamp. The set command from the client has 10 bytes. LabNet has to send two messages to indicate the input state. The client has also to send two messages to switch the output state. Additionally, LabNet sends two messages to acknowledge the output state switch. Thus, there is a total of six messages per iteration.

The results are summarized in *Figure 2b and d*. For C# the median was 0.89 ms for 4 and 1.23 ms for 2 and mean values were 0.9±0.19 ms for 4 and 1.24±0.18 ms for 2. For Python the median was 0.89 ms for 4 and 1.28 ms for 2 and mean values were 0.88±0.18 ms for 4 and 1.24±0.16 ms for 2. For C++, the median was 0.93 ms for 4 and 1.12 ms for 2 and mean values were 0.92±0.15 ms for 4 and 1.12±0.07 ms for 2.

LabNet uses a polling mechanism to detect changes in digital inputs. By default, LabNet runs at 4 kHz polling. But we also evaluated latencies for 1 kHz and non-stop polling on Pi 4. Mean values in case of 1 kHz were 1.23±0.16 and 0.75±0.13 ms for non-stop for C#. Four kHz with 0.9 ± 0.19 ms offers slightly worse results compared to non-stop polling, but on the other hand only utilizes 10% of the capacity of one CPU core.

As a further result, the compiler optimization flags did not influence any of the tests. This indicates that LabNet has no performance issues on the RasPi.

The 1 GigE update of the RasberryPi 4 causes a performance increase over models 2 and 3. Despite the differences in implementation, all clients are relatively close with their performance. The results also show that the language used at the client side is not important, at least for the simple cases considered here.

## Stress test

This is an extension of the 'read and set' tests. But now 14 pairs of pins were connected; all 28 GPIOs on both RasPis were used. The C++ program on the second RasPi ran up to 14 tests in parallel, each in an own thread. The pause between single measurement runs was set to 1 ms. We needed this pause to give all threads a chance to be executed. The C++ measurement program ran on the RasPi 3B+ as before and LabNet on 4.

The results in *Figure 2c* show that LabNet can monitor and control up to 14 pairs of IO-pins in parallel without any loss in performance. Interestingly, the latencies went down slightly for two test signals, but then remained at this level; the median for C# was 0.89 ms for one signal and 0.70 ms for two. This probably has to do with the polling in LabNet and the parallel test execution of the second RasPi. With two signals, it is more likely that LabNet will notice the pin state change within shorter delay.

Additionally, we looked at how many latency measurements per second the second RasPi could execute. With a single test signal this was just over 400 events per second. With the maximum of 14 tests each single pin switched only 200 but all pins together a total of 2800 times per second. The drop in the number of events per second from 400 for a single IO-pin occurs as soon as more than 4 IO-pin pairs are handled. This is a consequence of the four CPU cores on the RasberryPi. As soon as the test signal has been set, the software continuously monitors the state of the response pin. This keeps one CPU core fully busy and prevents the execution of the other test threads. However, this performance evaluation also shows that LabNet has no problems to process several thousand messages per second in each direction.

## Comparison

Our comparison of LabNet latency performance with other software tools is summarized in *Figure 3a*. We implemented an adapted version of the 'read and set' test for Autopilot, Bpod, and pyControl to achieve measurement comparability. The latency measurements were performed in exactly the same way as previously with LabNet. Two pins were connected to the measurement RasPi. The same C++ measurement program ran again on a RasPi 3b+, set the test pin to 1, and waited until the response pin was also set to 1. Tests were repeated 10,000 times. Different from LabNet all tools ran locally and did not send commands over the wire in the network. For source code and data, see the Code availability section.

In Whisker, the communication occurs over the network; however, both Whisker and all task clients usually run on the same PC. Such communication is extremely fast and is also reported by the Whisker authors (*Cardinal and Aitken, 2010*) to require only 0.066 ms. The 1 ms latency comes from Whisker's internal 1 kHz polling frequency for processing incoming commands. For Whisker we could not perform the 'read and set' test ourselves. Therefore, 1 ms is used as reference value.

Autopilot runs in a Rasberry Pi swarm. However, the tasks ran locally. To perform the 'read and set' test, the 'free water task' from the Autopilot GitHub repository was adapted. This waits for a digital input event, activates a digital output for a short time, and repeats. The measured mean latency was 0.93 ± 0.08 ms on RasPi 4.

The pyControl state machine is also very simple. It has only two states, which simply monitor the digital input and turn the digital output on and off. The mean latency is 0.66 ± 0.11 ms. This is comparable with reported results 0.56±0.02 ms from *Akam et al., 2022*. The used MicroPython pyboard version was 1.1.

Since the Bpod state machine runs at 10 kHz, we expected it to perform best which was the case. The mean latency was 0.1 ± 0.002 ms. We tested the version r2 of the Bpod State Machine.

According to these measurements, LabNet achieves latency times comparable to locally executed applications, even despite client control of LabNet over the network via TCP/IP.

## Discussion

With LabNet we present a C++ optimized control layer software to control Raspberry Pi connected hardware over a simple network protocol. LabNet can be used for general automation in experimental laboratories. And the controlling PC can be located at some distance. The version of LabNet presented here is not our first solution of distributed experimental hardware control. After initially using PC digital IO boards for 760 parallel IO lines (*Winter and Stich, 2005*), we moved to a custom developed microcontroller board connected to the PC initially via UART and later via Ethernet. In 2015 we switched to the Rasberry Pi avoiding own hardware development. We used a prior version of LabNet for 5 years before rebuilding over the past 2 years from the ground up the current highly optimized version of LabNet using our prior experience with laboratory experimental control. In the following we present some of the experimental systems that included LabNet control.

For our experiments with nectar-feeding bats we controlled a system of up to 76 artificial sugar water feeders (artificial flowers) each of which included a nose-poke sensor, an LED, a motorized swivel arm to close the flower and two valves for reward (*Winter and Stich, 2005*). Later the flowers were extended with RFID readers for the individual identification of ID chipped bats and has been used with freely ranging bats both in the rainforest (*Nachev et al., 2017*) and in the laboratory (*Wintergerst et al., 2021*). These systems had up to 23 flowers and each flower was accessible to all bat. While in

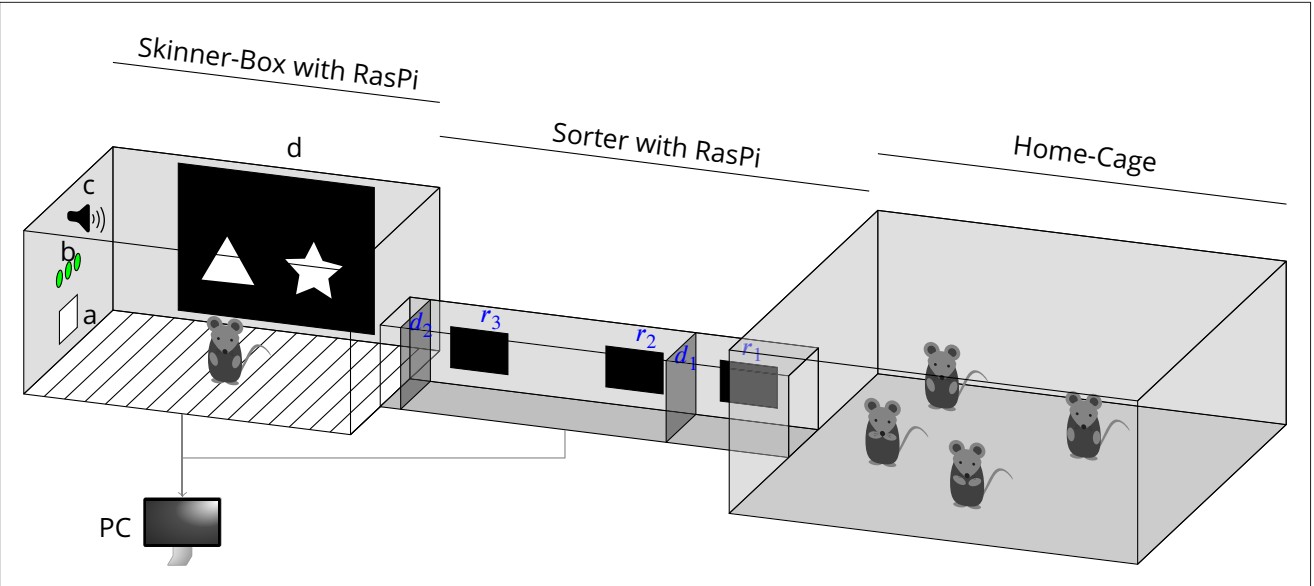

**Figure 4.** A complete behavioural setup controlled by LabNet. On the left is the touchscreen chamber. (**a**) Feeder magazine with a nose-poke sensor and a pellet dispenser, (**b**) a row of LEDs, (**c**) a tone generator, (**d**) monitor displays for visual stimuli with IR touch frame sensor. In the middle is the sorting module. ($r_1 - r_3$) three RFID readers. $r_2$ and $r_3$ are positioned so that the animals can be read anywhere inside the sorter and $r_1$ so that the animal is read when it leaves the sorter. (**d$_1$, d$_2$**) Two doors to catch the animal inside the sorter and guide it to the experimental chamber or the home cage. The animals live inside the home cage and can participate voluntarily and unsupervised in the experiments in the touchscreen chamber via the sorter. Both the touchscreen system and the sorter are each equipped with their own RasPi and connected to PC via Ethernet. The sorter can be removed without electrical adjustments, because it is controlled by its own RasPi and is therefore completely independent from the touchscreen system. Then experiments can be conducted with manually introduced animals.

the earlier systems we used a UART-to-Ethernet converter from Perle Systems to receive RFID data, we now use a custom 32-channel UART HAT for the Rasperry Pi with LabNet. Also the stepper-motor nectar pumps and the rest of the hardware (nose-poke sensor, LED, valves, etc.) are now connected to the RasPi and controlled via LabNet. As to the network, in the case of individually kept animals, each flower had its own RasPi while in flower fields, several flowers shared one RasPi, depending the distance between experimental units.

We also perform behavioural experiments with rodents. In a study on rational choice by mice, ID chipped mice in a group home cage could choose between four water dispensers built very similar to our artificial bat flowers, with nose-poke sensor, a valve, an RFID reader, and a syringe pump to deliver the water. Here, all hardware was connected to RasPi and controlled with LabNet. We have also used LabNet in connection with commercially available experimental chambers. An example is the touchscreen system for rats from Campden Instruments that we extended using LabNet that controlled a gating system (ID sorter) to automatically perform experiments with group housed rats (*Marion et al., 2017*) (see *Figure 4* and more below). The program for the sorting procedure on the PC started the experiment in the touchscreen chamber via a TCP/IP protocol implemented in collaboration with Campden Instruments every time a new animal was sorted in. This allowed us to conduct the experiments with multiple animals automatically and unsupervised.

More recently we have implemented an experimental touchscreen system for group housed mice that is fully under RasPi and LabNet control (*Figure 4*). This consists of a touchscreen system, a sorter, and a home cage. The touchscreen system has a monitor with an IR touch frame, a pellet magazine with pose-poke sensor, a row of LEDs, and a tone generator. All components are connected to a RasPi with LabNet control. Listings 2.2–2.2 show how this hardware can be initialized and controlled from the PC. The sorter has three RFID readers, two motorized doors, and two hall effect sensors, all connected to a RasPi. The readers connect via UART-USB converters, motors are controlled via UART, and the hall effect sensors for door state connect to IO ports. The sorting procedure, that is, when which door goes up or down, is realized by the PC, the client. The animals live inside the home cage and participate voluntarily and unsupervised in the experiments in the touchscreen chamber. *Figure 4* shows only one system, but we had up to four systems connected and controlled by one PC. This

also shows the advantage of a distributed system like LabNet. In order to control more systems, they were simply connected to the network without further adjustments. With identical systems, the same experiments could run everywhere at the same time.

LabNet is a very versatile distributed system which allows to control the hardware in laboratory and field experiments. It achieves almost real-time hardware control despite the network communication. Our stress test measurements have shown that thousands of Ethernet messages can be handled by LabNet per second. Indeed, the bottleneck here is the client and its ability to process and react to LabNet messages. However, none of our systems had reached the number of messages per second as in the stress test and we never had performance issues. The only problem could be very large messages in the network communication, for example, video data. These could significantly worsen the latency of other messages. But, there is the possibility to put RasPis with cameras into another network and on the PC receive the data via another network card. LabNet can also execute multiple tasks on one RasPi at the same time. In our experience with the touchscreen system we observed inputs, generated multiple pulse trains, played sound, and displayed visual stimuli, all at the same time but never reached performance limits on the RasPi. The LabNet architecture with actors explicitly targets the execution of multiple tasks.

Raspberry Pi as the hardware platform allows connecting a wide variety of readily available sensors and actuators. LabNet supports already a range of hardware modules which can thus be addressed via network. For example, GPIOs, communication via the UARTs, sound output via a headphone jack or HDMI, and some Raspberry Pi HATs developed in-house. This already allows many types of experiments. LabNet can also be used with hardware adaptors with available modules for operant experiments from open source such as Bpod and pyControl or commercial systems from MedAssociates or Coulbourn Instrumentsor. In addition, LabNet can be integrated into existing systems, as shown above with a Campden Instruments' system.

Here, we do not show how LabNet can be extended in C++ with new functionality. This is part of the API documentation which may undergo changes between versions and will be available online. The next version in progress will support the display of visual stimuli, touchscreen support, and communication via I2C. This version will also include a complete API documentation. The support for a Raspberry Pi-based configuration file is also planned. This would make configuration via the network no longer necessary and LabNet could already initialize the hardware correctly on Raspberry Pi start.

In the future, we plan to implement a software plug-in system. This will make it possible to support new hardware without LabNet recompiling. This will require a rework of the current LabNet API. This will then support messages that are unknown at LabNet compile time.

# Materials and methods

**Key resources table**

| Reagent type (species) or resource | Designation | Source or reference | Identifiers | Additional information |
|---|---|---|---|---|
| Software, algorithm | pyControl | https://github.com/pyControl/code.git | RRID:SCR_021612 | pyControl source code repository, v1.7.1 |
| Software, algorithm | Autopilot | https://github.com/auto-pi-lot/autopilot.git | RRID:SCR_021518 | Autopilot source code repository, v0.4.4 |
| Software, algorithm | Bpod | https://github.com/sanworks/Bpod_Gen2.git | RRID:SCR_015943 | MATLAB software for Bpod, Gen2 |
| Other | Bpod | https://www.sanworks.io/index.php | RRID:SCR_015943 | r2 Bpod State Machine |

*Continued on next page*

*Continued*

| Reagent type (species) or resource | Designation | Source or reference | Identifiers | Additional information |
|---|---|---|---|---|
| Software, algorithm | LabNet | https://github.com/WinterLab-Berlin/LabNet.git | SHA-1: 333bd58 | LabNet source code repository |

Data of the performance measurements, the source code for Autopilot, Bpod, and pyControl tasks and the source code for the graphs are included in the article's data and source code repository.

## Actor model

Developing a system with multiple threads still requires much care and can be challenging. Thread local state and program global state have to be protected. Some type of locking mechanism is required. Unfortunately, the locking mechanism itself increases not only the scalability but also the code complexity and error-proneness due to the locking order. Locking problems such as race conditions or dead-locks must be avoided. But time and execution order-dependent errors can be difficult to find and fix.

LabNet is a concurrent system. The operations on the GPIOs, sending and receiving data via UARTs, sound output, etc. have to be independent from each other. For such purpose, message-passing approaches have been developed. In those, all inter-thread state sharing is encapsulated within messages sent between threads. All messages must be immutable or be copied for each thread.

Hewitt, Bishop, and Steiger (*Hewitt et al., 1973*) proposed in 1973 with their actor model one of the first message-passing systems. Actors are active objects that communicate only over messages. Each actor has only knowledge about itself and its own functioning (shared-nothing principle). No global state exists in an actor system. Messages also do not block the sender (fire-and-forget principle). This avoids problems such as race conditions or dead-locks.

This has further developed to a level of abstraction from only considering shared memory to independent actors that communicate through a well-defined message protocol. In the late 1980s, Ericsson developed Erlang (*Armstrong, 1996*), an actor-based programming language, and successfully used it in ATM network switches. The Akka (*Lightbend, 2021*) actors framework was released in 2009 for Java and Scala.

## SObjectizer

From the several actor model libraries that are available for C++ such as the C++ Actor Framework (CAF) (*Charousset et al., 2013*), SObjectizer (*Stiffstream, 2021*), or Theron (*Mason, 2019*) we chose SObjectizer.

In SObjectizer a class or struct is sufficient to define a message. Actors are also normal classes derived from an agent_t base class. Thus, actors automatically have a 'message box' (Mbox), through which messages can be received, and also methods that are automatically called, for example, before an actor is started or stopped.

An Mbox can receive messages of all possible types. The Mbox of an actor has no name and must be communicated to other actors. However, named Mboxes can also exist. A reference to such an Mbox can be created anytime via its name. This practical feature is used in LabNet to access important actors that always exist.

It is also possible to mix actors with other paradigms. This allows to move some parts of the application into the Boost ASIO (*Boost, 2021*) or into threads. Mboxes can still be used for communication with actors from the outside. For communication with code from outside the actor world, the so-called MChain are used. An MChain looks like a queue: actors can place messages there and threads can pull them at a later time. For example, LabNet uses Boost ASIO for TCP/IP communication and threads for some clearly defined tasks such as digital input polling.

One important feature of SObjectizer is the built-in support for hierarchical state machines (HSMs). All actors in SObjectizer are state machines. They can pass through several states in their lives and

react to incoming messages depending on their current state. An interface actor in LabNet (see Implementation section) goes through several states: hardware initialization, operation, error, etc.

Dispatchers are another important cornerstone of SObjectizer. Dispatchers provide an actor with the working context. They manage all message queues and execute the actors if there are messages in the queue (Mbox). We have chosen the dispatcher with a thread pool. It provides a good compromise between thread management overhead and parallelism. But it is still possible to use other dispatcher types (e.g. one with one thread per actor) without having to adapt the actors.

## Message protocol

Our criteria for choosing the serialization tool were good performance, small message size, and support in as many programming languages as possible. Text-based serialization formats, such as XML, JSON, or ASCII-based plain-text, have the advantage of being human-readable. The Whisker server uses an ASCII-based format (*Cardinal and Aitken, 2010*). The disadvantages are the message size, higher computing requirements, and, at least for the ASCII version, a custom message parser.

For our application, a binary format is a better solution, and we chose Protobuf *Google, 2021*. It is very popular and offers support for many programming languages. However, Protobuf has some disadvantages. For example, it is not the most memory or computationally efficient tool. Libraries such as Flatbuffers, Cap'n Proto, or Simple Binary Encoding (SBE) are more efficient. However, these negative aspects of Protobuf only become critical when sending extremely large messages (some MBytes) or at a very high rate (millions per second) . This is typically not the case in experiments that focus on actions of animal behaviour.

Protobuf uses a special meta-language to define messages. With protoc-generator, it is possible to create these message protocols for each supported programming language. Files with message definitions are a part of the Git repository.

One Protobuf disadvantage must be mentioned. A serialized Protobuf message contains no information about the byte length nor the message type. Protobuf leaves this information to the transmission medium. We have solved this simply: each message begins with two pieces of information: type and size. This is also the officially recommended approach. Both are encoded as a number in Protobuf's varint notation and are easy to parse with the Protobuf API.

## Implementation

The current implementation does not contain configuration files for LabNet. The hardware initialization is exclusively performed through client messages. LabNet comprises several loosely coupled actors. The most important are briefly described below (see also *Figure 5*).

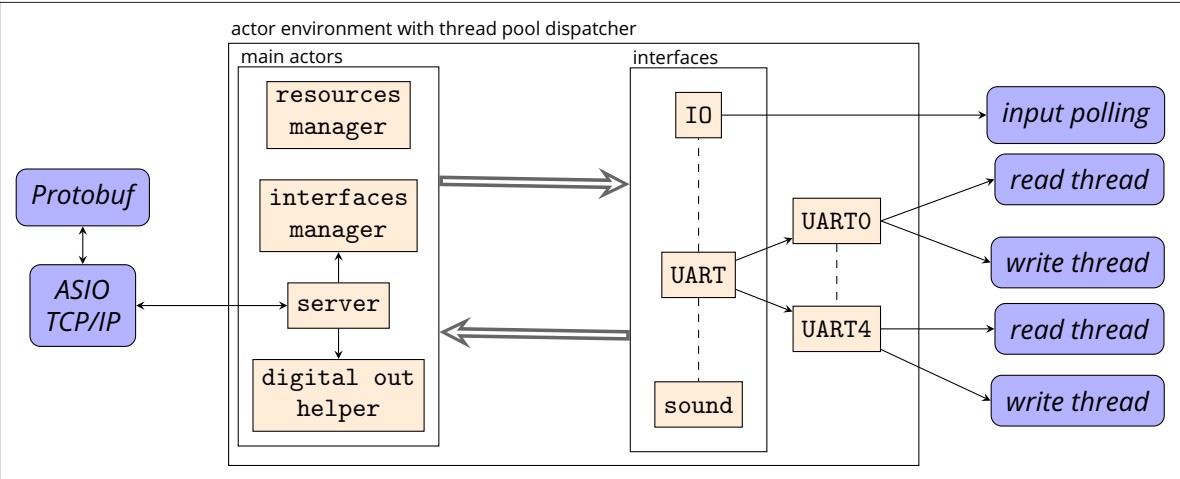

**Figure 5.** The core of LabNet is the actor environment with thread pool dispatcher. Within the environment, the main actors are always present. The actors of the individual interfaces are started by the interfaces manager as needed. The interface actors can also outsource their work to other actors and threads. All main and interface actors can communicate with each other. The threads themselves and network communication via Boost ASIO, on the other hand, are hidden behind their actors.

The network communication runs over TCP/IP. The server in LabNet is implemented in Boost ASIO (**Boost, 2021**). The implementation is hidden behind the server actor. This actor can send and receive the Protobuf messages and also informs the actor world about the connection state. If the connection is lost, the actors can stop their work and automatically continue it later at the same point on a reconnection.

At the beginning, no single interface actor to communicate with the hardware exists. These actors combine all the possibilities for the hardware control: for example, initialization, set or get digital pin state, etc. They are automatically created and started by the interface manager. Currently, several 'interfaces' exist:

1. GpioWiringPi to control input and output pins with WiringPi (**Henderson, 2019**).
2. IoBoard is a self-developed PCB top plane with power supply and pin connectors.
3. UART can send and receive data over the internal RasPi UART and USB to RS232 converters.
4. UartBoard is a self-developed PCB top plane with up to 32 UART connectors using SPI.
5. Sound allows a simple sound output in the form of sine tones over HDMI or a headphone jack.
6. BleUart is a Nordic UART service over Bluetooth Low Energy (BLE) and allows to communicate with Bluetooth devices.

Many pins on the Raspberry Pi offer more than one functionality. Clear responsibility for a hardware resource must be ensured. Each 'interface' actor must request the resources from the resource manager. This is one of the first steps during the interface initialization state.

The 'interfaces' with digital outputs offer only the possibility to switch the output pin state. More complex procedures are implemented by the digital out helper actor. This actor can automatically turn off a pin after a defined time or generate pulses by specifying the on/off duration and number of pulses. Additionally, a group of pins can be automatically switched on and off together in a loop.

## Related work

Most of the comparable software control tools published for behavioural experiments are more general packages. In addition to hardware control, they offer a more or less powerful tool for creating experiments, a user interface and a possibility to visualize the data. Although LabNet is only responsible for the hardware, a comparison is still worthwhile.

### Wisker-Server

The development of Whisker control suite started in 1999 by Cardinal and Aitken at the Department of Experimental Psychology, University of Cambridge, and is ongoing (Cardinal and Aitken, 2010). Initially, the aim was to use the existing resources of a PC and plugged-in IO cards to control behavioural experiments with visual stimuli and touchscreens in several boxes simultaneously. This was solved by an additional software layer where Whisker operates as the server and controls the hardware. The clients must connect to the server over TCP/IP, and each one controls an experiment in one of the chambers. The clients themselves can be written in any programming language. Communication occurs through a plain-text protocol.

Because of the outsourcing of the experiments to the clients, Whisker's approach is similar to ours. Due to the flexibility in implementing the clients, complex experiments can be realized with Whisker. Hardware support includes digital I/O devices (National Instruments, Advantech, etc.), visual stimuli on computer monitors, touchscreens, audio, and more. Whisker is commercially used in 'ABET II' by Campden Instruments Ltd.

### pyControl

pyControl (**Akam et al., 2022**) is an open-source hardware and software framework for controlling behavioural experiments. The hardware is based on the MicroPython microcontroller that typically controls a single experimental box each. Several pyControl breakout boards can connect to a PC via USB. Each board has six so-called behaviour ports and four BNC ports. Each port can be connected to a module: to drive LEDs, nose-poke sensors, stepper motors, and speakers. Two behaviour ports have I2C internally and can drive a port expander module to increase the number of ports.

Tasks on the MicroPython microcontroller and pyControl on the PC use Python. A task is defined as a finite-state machine. It comprises a collection of states and events that cause the switch between states. In data management, all events and state changes are stored with timestamps.

pyControl provides sufficient I/O ports to realize most tasks on a system. However, for the hardware types, we are limited to the firmware capabilities and available modules, although free wiring is also possible. The mandatory requirement to define the task as a state machine can be useful but may also become a limitation.

### Bpod

Bpod *Sanders, 2021* was originally developed in the Brody lab and is now maintained by Josh Sanders (Sanworks LLC.). It also has been expanded to PyBpod as a python port of the Bpod MATLAB project by members of the Champalimaud Foundation. Bpod offers only four I/O ports but has additional module ports that each provide an interface to Arduino-powered modules. Thus, Bpod gains additional flexibility: analog I/O, I2C, Ethernet, and more can be accessed via these modules.

A MATLAB package is offered to write experimental tasks. Unfortunately, the package documentation is limited. The tasks are also defined as finite-state machines. After starting the task, the state machine is transferred to the Bpod. From there, it communicates with the MATLAB frontend. This design results in the restriction that only a single Bpod can be controlled per MATLAB session. Therefore, Bpod is much more limited regarding software than pyControl or Whisker. Multiple systems cannot run simultaneously, and the functionality is limited by the firmware and the state machine.

### Autopilot

Autopilot *Saunders and Wehr, 2019* is an open-source framework for behavioural experiments developed in the Wehr Lab at the University of Oregon. It uses Python, and the target platform is the Raspberry Pi.

The focus of Autopilot from the beginning has been the ability to control multiple systems. The basic unit in the software architecture of Autopilot is an agent. Each agent runs on its own Raspberry Pi and can communicate with other agents. Currently, three types of agents exist: terminal, pilot, and child.

Terminal agents are the only user-oriented with a graphical user interface. They are responsible for data logging and visualization. The experimental tasks are also managed here and transferred to the pilots, which are also responsible for experimental task execution. The pilots communicate with the external hardware that is connected to the Raspberry Pi and forward the experimental data to the terminals for logging or visualization. Each pilot can also have several child agents. Child agents can take over a part of a task if the task has been configured accordingly. The child agents are invisible to the terminals and communicate only with their parent pilot.

Among all tools discussed here, Autopilot offers most flexibility. It already supports a whole range of hardware. This includes digital I/O, audio, cameras, and some sensors such as temperature. Moreover, since it is open-source, support for additional hardware can be added. New behavioural experiments can also be implemented. However, in both cases, we are limited to Python.

## Code availability

The source code of LabNet is available over the GitHub repository under a GPL-3.0 license.

Data of the performance measurements, the source code for Autopilot, Bpod and pyControl tasks, and the source code for the graphs are also accessible via the GitHub repository, (copy archived at swh:1:rev:d52e52c51e3f7c5b0e12f95829b8cf4886bb3379; *Schatz and Winter, 2022*).

There are instructions for two possible compilation paths. The first is on the Raspberry Pi with Visual Studio Code and CMake. The second is with Visual Studio 2019 and Docker. The archive also contains the source code of all tests from Performance evaluation under 'examples'.

## Acknowledgements

We thank R Cardinal, T Akam, J Sanders, and J Saunders for comments on an earlier version of the manuscript.

Support for this work was received through the Deutsche Forschungsgemeinschaft (DFG, German Research Foundation), SFB 1315, project-ID 327654276, and EXC 257: NeuroCure, project-ID 39052203.

## Additional information

### Funding

| Funder | Grant reference number | Author |
| --- | --- | --- |
| Deutsche Forschungsgemeinschaft | SFB 1315 project-ID 327654276 | Alexej Schatz |
| Deutsche Forschungsgemeinschaft | EXC 257: NeuroCure project-ID 39052203 | Alexej Schatz |

The funders had no role in study design, data collection and interpretation, or the decision to submit the work for publication.

### Author contributions

Alexej Schatz, Software, Visualization, Writing - original draft; York Winter, Conceptualization, Supervision, Funding acquisition, Methodology, Writing - review and editing

### Author ORCIDs

Alexej Schatz ⓘ http://orcid.org/0000-0002-2664-2103

### Decision letter and Author response

Decision letter https://doi.org/10.7554/eLife.77973.sa1
Author response https://doi.org/10.7554/eLife.77973.sa2

## Additional files

### Supplementary files

• Transparent reporting form

### Data availability

Tool source code and performance measurements are available on GitHub (https://github.com/WinterLab-Berlin/LabNet and https://github.com/darki-31/LabNet_manuscript_data respectively).

The following dataset was generated:

| Author(s) | Year | Dataset title | Dataset URL | Database and Identifier |
| --- | --- | --- | --- | --- |
| Schatz A | 2022 | performance measurements data | https://github.com/darki-31/LabNet_manuscript_data | GitHub, LabNet_manuscript_data |

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
