## [Editor Report]

LabNet is an exciting new platform for experimental control using Raspberry Pis. As experiments get more complex in neuroscience, new validated tools are needed to continue to allow users flexibility, precision, and be fast, and LabNet is such a tool. Through extensive benchmarking and documentation of their tool, they demonstrate excellent performance, scalability, and provide examples of how their platform can be adopted.

---

## [Decision Letter]

**Decision letter after peer review:**

Thank you for submitting your article "LabNet: hardware control software for the Raspberry Pi" for consideration by *eLife*. Your article has been reviewed by 2 peer reviewers, and the evaluation has been overseen by a Reviewing Editor and Kate Wassum as the Senior Editor. The following individuals involved in the review of your submission have agreed to reveal their identity: Jonny L Saunders (Reviewer #1); Gonçalo Lopes (Reviewer #2).

Essential revisions:

Alexej Schatz and York Winter wrote "LabNet," a C++ tool to control Raspberry π (raspi) GPIO (General Purpose Input-Output) and other hardware using a network messaging protocol using protobuf. The authors were primarily concerned with performance, specifically low execution latencies, as well as extensibility to a variety of hardware. LabNet's network architecture is asymmetrical and treats one or many raspis as servers that can receive control signals from one or more clients. Servers operate as (approximately) stateless "agents" that execute instructions received in message boxes using a single thread or pool of threads. The authors describe several examples of basic functionality like time to write and read GPIO state to characterize the performance of the system, the code for which is available in a linked GitHub repository.

Overall, the described performance of LabNet is impressive, with near- or sub-millisecond latency across the several tests when conducted over a LAN TCP/IP connection. The demonstrated ability to interact with the server from three programming languages (C++, C#, and Python) also would be quite useful for a tool that intends to be as general-purpose as this one. The design decisions that led to the use of protobuf and SObjectizer seem sound and supportive of the primary performance goal. We thank the authors for taking the initiative to contribute more open-source tools to the community, and for addressing these hard challenges that keep recurring in systems neuroscience experiments. We absolutely need more people working on this in an open way.

We do ask the authors for the following revisions:

Technical:

– The main method for evaluating the performance of LabNet is a series of performance tests in the Raspberry π comparing clients written in C++, C# and Python, followed by a series of benchmarks comparing LabNet against other established hardware control platforms. While these are undoubtedly useful, especially the latter, the use of benchmarking methods as described in the paper should be carefully revisited, as there are a number of possible confounding factors.

– For example, in the performance tests comparing clients written in C++, C# and Python, the Python implementation is running synchronously and directly on top of the low-level interface with system sockets, while the C++ and C# versions use complex, concurrent frameworks designed for resilience and scalability. This difference alone could easily skew the Python results in the simplistic benchmarks presented in the paper, which can leave the reader skeptical about all the comparisons with Python in Figure 3. Similarly, the complex nature of available frameworks also raises questions about the comparison between C# and C++. I don't think it is fair to say that Figure 3 is really comparing languages, as much as specific frameworks. In general, comparing the performance of languages themselves for any task, especially compiled languages, is a very difficult topic that I would generally avoid, especially when targeting a more general, non-technical audience.

– The second set of benchmarks comparing LabNet to other established hardware control platforms is much more interesting, but unfortunately, it doesn't currently seem to allow an adequate assessment of the different systems. Specifically, from the authors' description of the benchmarking procedure, it doesn't seem like the same task was used to generate the different latency numbers presented, and the values seem to have been mostly extracted from each of the platform's published results. This reduces the value of the benchmarks in the sense that it is unclear what conditions are really being compared. For example, while the numbers for pyControl and Bpod seem to be reporting the activation of simple digital input and output lines, the latency presented for Autopilot uses as a reference the start of a sound waveform on a stereo headphone jack. Audio DSP requires specialized hardware in the π which is likely to intrinsically introduce higher latency versus simply toggling a digital line, so it is not clear whether these scenarios are really comparable. Similarly, the numbers for Whisker and Bpod being presented without any variance make it hard to interpret the results.

Documentation

– Could the authors provide some example code and minimal working examples such that the average user could easily jump in?

Currently, LabNet has no documentation to speak of, outside a brief description of the build process for a relatively voluminous body of code (~27k lines) with relatively few comments. There is no established norm as to what stage in a scientific software package development a paper should be written, so I take the lack of documentation at this stage as just a sign that this project is young. The primary barrier for the broader landscape of scientific software is less that of availability of technically proficient packages, but the ease with which they can be adopted and used by people outside the development team. The ability of downstream researchers to use and extend the library to suit their needs will depend on future documentation. For example, at the moment the Python adapter to the client and server is present in the examples folder but relatively un-annotated, so it might be challenging to adapt to differing needs at the moment (https://github.com/WinterLab-Berlin/LabNet/blob/34e71c6827d2feef9b65d037ee5f2e8ca227db39/examples/python/perf_test/LabNetClient_pb2.py and https://github.com/WinterLab-Berlin/LabNet/blob/34e71c6827d2feef9b65d037ee5f2e8ca227db39/examples/python/perf_test/LabNetServer_pb2.py). Documentation for projects like this that aim to serve as the basis from which to build experimental infrastructure can be quite challenging, as it often needs to spread beyond the package itself to more general concerns like how to use Raspberry Pis, how to set them up to be available over a network, and so on, so I look forward to seeing the authors meet that challenge.

Manuscript:

– It also would be worth commenting more on the intended mode of use of LabNet. The authors do this a bit in the introduction setting the scope of the package as performant GPIO control rather than full experimental control, but they might consider expanding a bit on how they intend it to be used to run experiments. The model of having one computer that runs the experiment and another that executes the hardware control is more similar to Bpod than Autopilot or PiControl, both of which are intended to have the raspi/micropi be autonomous (so the network latency for issuing commands is less relevant because you would put time-critical operations on a single π or else hardwire them with a GPIO pin), and this comparison is useful for understanding the broader landscape of experimental control software in my opinion.

– Some of the technical explanations of the choices of software libraries are unclear to me and I had to do a decent amount of additional reading to understand them. I'm still not exactly sure what SObjectizer does. The same is true of the description of the actor model – a bit of history is given, but I think more of a description about why qualities like statelessness are valuable, what the alternatives are, and a figure that clarifies the concurrency/agent model would be very useful. I recognize this is challenging because it's not altogether clear what you can expect from your audience, but the manuscript as written leaves me a bit unclear at the internals and I had to go read the code.

Please provide a key resource table, if you have not already done so.

*Reviewer #1 (Recommendations for the authors):*

First, I want to thank the authors for their work! I put most of my positive comments (and most of them were positive!) in the public review section, so my apologies if this section is mostly recommendations rather than praise.

The comparison in Figure 4 compares a reasonably broad range of functionality: the reported LabNet measurements are for the latency from sending an output signal on one pin to reading it on another. PyControl is the closest (latency from reading an input to writing an output), but the Bpod latency is from a software command to the onset of sound output, and the Autopilot latency is from a hardware input to the onset of sound output, both of which are different tests in substance because of the typical need to write audio output in frames rather than as scalar values which have intrinsic latency. Relying on reported values is ok, but the interpretation in the text lacks some clarifying context for the comparison in Figure 4, and in my opinion, the authors would be well served by running these tests themselves on the other systems (in the case of Autopilot, this would require purchasing no additional hardware). This leads some of the descriptions in the text to be inaccurate: for example, a comparable test to the "set and read GPIO" test in Autopilot takes actually ~2.3ms rather than the 12ms the authors estimate (see: https://gist.github.com/sneakers-the-rat/41683e42da73712277c355dfa612af96), and setting and checking a digital output locally takes ~.14ms +/- 0.05 rather than the reported 1.75ms. To be clear, I don't see this as a major problem with the manuscript, since it is not a problem with the software in question, but with reporting of prior results.

It also would be worth commenting more on the intended mode of use of LabNet. The authors do this a bit in the introduction setting the scope of the package as performant GPIO control rather than full experimental control, but they might consider expanding a bit on how they intend it to be used to run experiments. The model of having one computer that runs the experiment and another that executes the hardware control is more similar to Bpod than Autopilot or PiControl, both of which are intended to have the raspi/micropi be autonomous (so the network latency for issuing commands is less relevant because you would put time-critical operations on a single π or else hardwire them with a GPIO pin), and this comparison is useful for understanding the broader landscape of experimental control software in my opinion.

Some of the technical explanations of the choices of software libraries are unclear to me and I had to do a decent amount of additional reading to understand them. I'm still not exactly sure what SObjectizer does. The same is true of the description of the actor model – a bit of history is given, but I think more of a description about why qualities like statelessness are valuable, what the alternatives are, and a figure that clarifies the concurrency/agent model would be very useful. I recognize this is challenging because it's not altogether clear what you can expect from your audience, but the manuscript as written leaves me a bit unclear at the internals and I had to go read the code.

A point that I raised a bit in the public review is that I think it's important to weigh the different practical considerations of experimental code: LabNet's performance seems great, but how does that trade-off with using C++, a language that in my experience far fewer scientists know how to use than Python? This called out to me especially strongly considering that in several of the performance tests the Python client seemed to be faster than the C# or C++ clients (Figure 3a and c, RV4) – for what it's worth this is how Autopilot works, controlling a low-level process from Python. Apart from a discussion of these tradeoffs, that should be addressed specifically because the framing of the paper suggests that C++ should always be faster. I am also not sure what 3d adds, and in any case, both set and read gpio results should be presented sequentially (ie. 3b should be set test, 3c should be set and read gpio medians, 3d should be means with stdevs) so that the values can be compared between the two metrics on the same test.

I'm not really sure how the listings relate to the library code, as the examples in the github repository seem considerably more complicated than presented in the text. If these are intended to be illustrative, that should be stated clearly. I think that some inline comments to clarify the finer points of the code (eg. what does `Interface = GpioWiringpi` do? how is that different than `GpioWiringPiInitDigitalIn` ?) would also be useful.

As I noted in the public review, I think it's worth discussing the state of the library, in particular, any plans for documentation – without docs, it looks like it would be quite intimidating to adopt, but a note clarifying that they are in progress, etc., might soothe nerves.

I was unable to run the tests myself, and I raised a few issues with compilation and use (https://github.com/WinterLab-Berlin/LabNet/issues/2 and https://github.com/WinterLab-Berlin/LabNet/issues/1) but didn't want to hold up the review any longer trying to get it to work – I have no doubts that the code does what the authors describe and have no reason to doubt their results.

All in all, thanks again for your work, hopefully, these suggestions are useful!

*Reviewer #2 (Recommendations for the authors):*

As mentioned in the public review comments I fully subscribe to the two stated goals for LabNet mentioned in the paper, and I think most of the improvements for future revisions should focus on discussing them both more thoroughly. I would frame the entire Design discussion around these two topics, rather than on strictly the Actor-model. Indeed, the Actor model should feel more like a means for you to achieve your stated aims, and specifically, I think it would be great if you could clarify how exactly it helps to achieve both time-critical operations and ease of extending the system.

L52-53: The way this last sentence is phrased somehow gave me the misleading impression that a server would support more than one client, which made me even more surprised to read L161.

L56-L58: It is stated that the Implementation section will provide implementation details to compare LabNet with other systems, but it's not clear how any of these details are important for the benchmarks. It might be just an issue with phrasing.

L65: The discussion on how to develop new functionality extensions seems to be entirely missing, and to me, this was one of the big surprises when I finished reading the paper. Could it be you were thinking purely about combinatorial flexibility, i.e. combining existing functionality to create a new experiment? This is not clear, and right now it is really compromising the entire manuscript, especially since at the end of the Conclusions section you seem to imply that LabNet is not ready to easily accept modifications without recompiling everything from scratch.

Figure 1: When seeing the example applications, the first thing that jumps to the forefront is no mention of video anywhere. Also, these applications should really be more detailed in later sections. Are these already working in LabNet? If so, they would make great examples to illustrate how exactly the components are combined to make a real-world experiment. If there is any kind of preliminary data that could be published, this would of course really strengthen the manuscript.

L71-72: This sentence is very confusing, but I guess the point was to emphasize the existence of asynchronous systems in the Pi?

L3: "fast" instead of "rapid".

L76: Reference to actor-based models should be first introduced here.

L80-L85: This whole bit on locking seems unnecessary. It is not that locking is not important anymore, but the way you introduce it currently is confusing as it is not clear how any of this relates to LabNet specifically.

I think the end of the Actor model section talks about too many topics which are never clearly connected back to LabNet, such as CSP. If they are indeed connected, this should be made explicit in the text, and the relevance of the connection to the argument should be presented. Otherwise, they should be removed.

L119: The reference to Boost ASIO should be introduced here, to avoid confusing unfamiliar readers.

L123-125: The relevance of the built-in support for hierarchical state machines is unclear. Elsewhere it is mentioned that it is not intended for LabNet to implement complex state logic in the nodes, so it is not obvious whether this support would be useful for anything. If it is, it should be made explicit.

L126-134: It seems like LabNet currently only uses the default dispatcher, so I would condense this paragraph and move the details to the software documentation pages.

L145-146: How large is "extremely large" (e.g. in bytes) and how fast is "a very high rate" (e.g. in Hz)?

L152: Given there are custom modifications to Protobuf, it might be worth spending some time describing how exactly messages are encoded, and what kind of information is transmitted by device events and commands.

L157: It seems the client must know beforehand what devices are connected to each Raspberry Pi, and in what pins. If so, it would make the presentation clearer to list what are the available devices and their characteristics.

L162: It is not clear to a reader unfamiliar with the underlying implementation what is the issue with clients accessing "foreign hardware". It would be important to clarify.

L165: What, if any, state is recovered when actors stop and resume their work?

L168: What exactly is an "interface"? This point needs more elaboration to allow a reader to understand the underlying LabNet architecture in detail.

L193: Isn't the sine tone defined in Listing 1 rather than Listing 2 where it seems to be a digital square pulse? The purpose of each example in the list is unclear.

Listing 1 and 2. The switch to using C# would benefit from a bit more clarification and explanation. I understand the platform supports clients to be written in any language, and this might be an example of that fact, but it would still be beneficial to clarify that these examples are not running on the Pi, but rather in the client software which talks to the π instead.

L277: It is strange to benchmark python as a stripped client running directly on top of the sockets when both the C++ and C# clients make use of much more complex frameworks handling a variety of concerns. Specifically, I don't think it's fair to call this a comparison between "languages", since neither SObjectivizer nor Akka.NET forms part of C++ or C#, not even part of their standard libraries. Also, C# has now moved on to.NET Core for cross-platform server and network implementations, with presumably much-improved performance.

L311: How do you distinguish here whether the increase in latency is due to "C#" as opposed to some implementation detail in Akka.NET?

L344: Did you reproduce this test on LabNet running the audio headphone jack output? Just to make sure the DSP output is exactly the same and exclude all other possible forms of latency?

L361: This claim was not investigated in the manuscript, since no example of an actor implemented in SObjectivizer was given. See public review comments.

L370: How are visual stimuli defined in LabNet? General stimulus display frameworks are notorious for developing intricate dependencies and it would be a great exercise to include how you are thinking about composition in that case.

L375: These limitations seem to go against the 2nd goal of LabNet?

[Editors’ note: further revisions were suggested prior to acceptance, as described below.]

Thank you for resubmitting your work entitled "LabNet hardware control software for the Raspberry Pi" for further consideration by *eLife*. Your revised article has been evaluated by Kate Wassum (Senior Editor) and a Reviewing Editor.

The manuscript has been improved but there are some remaining issues that need to be addressed, as outlined below:

Overall, some of the more critical revisions we feel were not adequately addressed. The reviewers and I have consulted, and I agree that the points below need to be addressed for the manuscript to be suitable in *eLife*. To summarize required major revisions:

– Remove language about other packages not meeting their claims re: stress test (as they were not tested).

– Clarify wording on stress test results (see R#2 review).

– Describe means of measuring timings on both LabNet and other packages.

– Link to specific versions of code for each of the tests.

– Describe versions (with git hashes or semver) of all software, LabNet and other packages.

– Respond to R#1 questions about example tests for other packages.

– Describe an example experiment done within the lab using LabNet and how it fits into the rest of the experimental setup.

– Label the axes in all figures.

– Clarify the benchmarking protocol well, with a diagram.

– Please include a key resource table.

*Reviewer #1 (Recommendations for the authors):*

General comments:

– Many of these will read as negative, so I want to start by saying I appreciate the author's work, apologize for the lateness of my review (life has been hectic pre- and post-dissertation!), and thank them for writing this package! All that I don't comment on here I think is good.

– In general I like to see software timestamp measurements supplemented with hardware measurements (from eg. an oscilloscope), even just to confirm that the software timestamps are close. I don't think it's of huge importance here, but I wanted to make that future recommendation to the authors, especially when taking timestamps from an interpreted language like Python.

– The mismatch between L84-L88 and the results is made more salient with the addition of L143-L147 – L84-88 say Python is intrinsically slow and thus C++ was chosen, but then L143-147 say that Python has an advantage because the C++ implementations are more complex. L84-88 thus read like theoretical concerns that were demonstrated here to not be true because of additional details in the C++ implementation, right?

– I appreciated the expanded discussion of the intended uses for the package, like the discussion of the potential for using multiple pis together, etc. I think that and the brief descriptions of potential tasks help the paper!

– I don't see a discussion of documentation in the main text, I don't think it's worth holding the paper up over, but I again make the recommendation to the authors to at least discuss their plans for documentation and future maintenance, as that is really the critical factor for whether a package like this will be adoptable by other labs. The authors briefly address this in their response, but yes this is important information for prospective users to have!

– Some of the other concerns that I raised in the prior recommendations for the authors were not addressed, perhaps that was my fault in not understanding how the public review. vs recommendations to authors work at *eLife*.

Figure comments:

Additional comments on new text:

– Listings: The inline comments and in-text descriptions are much appreciated!

– Figure 1: You designed that in TikZ? I am amazed. I would love to see that code. I checked out the TikZ code for the other figures and am very impressed.

– Figure 2b-d: The y-axes are unlabeled.

– L17-18: I don't see stress test comparisons for the other packages, so the "unlike others" doesn't seem to be supported by the text.

– L18: typo, latenies -> latencies.

– L61-63: This seems like an odd definition of openness to me, which typically means either that the source is inspectable. I would call the "control an experimental chamber on its own" part independence or modularity, and the "or together with a number of other nodes" interoperability or scalability. I am unsure how one would use multiple LabNet nodes in the same task, as an example doesn't seem to be in the text! This also seems to contradict L66-67 "However this comes with the restriction that at most one experimental system can be connected to each RasPi" – what counts as an experimental system here? are the authors just referring to a particular set of hardware components which could be combined in a single experimental chamber? that clarification would resolve the conflict to me.

– L71-72: I am not sure what this means, the client is the controlling computer, but not sure what a task is in this context. And I thought that the hardware control happened on the raspi (server?).

– L72-73: from what I recall you also provide clients in these languages? might be worth some clarification describing what you mean by writing clients in multiple languages – eg. that clients written in multiple languages can interact with the underlying C++ library?

– L171: I found the description of the new crossover-based tests a bit hard to follow and it took me a few reads, I think a diagram would be helpful here :

– L210-221: I can't really tell how the code works for the new tests, it would be really helpful to link to the source code for each of the tests in the main text so we know what you're referring to! for example, the stress test looks like it does not send instructions of the network but operates on local logic as well: https://github.com/WinterLab-Berlin/LabNet/blob/3963f3371610d828e44af1e27ba6374cacc79748/examples/cpp/perf_test/main.cpp

I see the data availability statement and found the repo, but it would be nice to have those separated out instead of inside of a zip file so you could link to them.

– L228-229: The inclusion of whisker now reads as odd, since I think assuming latencies based on polling frequency is probably a pretty bad assumption in most cases.

– L218-241: I am not sure how the latencies are measured for the other systems, as it looks like you were taking software timestamps for the LabNet tests, but I don't see timestamps being taken in any of the other comparison tests, so I assume that external timestamps were being taken? It also seems like some external trigger would be needed in some of these tests as well (see below)? It is also important to validate any software timestamps with external hardware timestamps, and they shouldn't be mixed without validation (eg. software timestamps could either exaggerate or underestimate latencies depending on where they are taken). Some additional clarification is needed here.

From my reading it doesn't look like the pycontrol or autopilot tests would work, but I am out of the lab and don't have a π to run them on myself.

For the pycontrol test, it seems like it would go

- on_state(event='entry') -> p_off

- off_state(event='p_off') -> goto on_state

- on_state(event='entry') …

and if the 'on' state was triggered manually from an external trigger it would go like

- on_state(event='p_on') -> goto off_state

- off_state(event='entry') -> switch p_out.on()

- on_state(event='p_on') -> goto off_state

- off_state(event='entry') …

But I admit I am not altogether familiar with pycontrol. Some comments in the source would be lovely.

For the Autopilot test, there are two stage methods in a cycle, water() and response(). the water method clears the stage_block which would cause the running pilot to wait for any set triggers. A trigger is set for the 'sig_in' pin that should set the 'sig_out' pin high. When the sig_in trigger is received, that should cause the response() method to be called which sets 'sig_out' low after some delay and then return to the water stage immediately. This would require some external input to trigger the sig_in event, and then the timestamps of the external input and the sig_out pin would be compared. If the sig_out pin were wired to sig_in, the test wouldn't work, as the sig_out pin would never be set high. Having the `prefs.json` file from the pilot would be useful to include here to avoid ambiguity in system configuration, as I am assuming the default polarity configuration settings are used on the digital input and output classes.

A set of tests that are more similar to the tests described for labnet are available in the plugin accompanying our manuscript: https://github.com/auto-pi-lot/plugin-paper/blob/c6263a4890b7d6101688158d8acb3aaeb9199533/plugin_paper/scripts/test_gpio.py documented here: https://wiki.auto-pi-lot.com/index.php/Plugin:Autopilot_Paper We find the roundtrip (oscilloscope measurement of input to output) latency to be 400us.

I don't doubt the authors ran the tests, and please correct where my read of the code is wrong, but I think some additional detail is needed in the reporting of the results in any case.

*Reviewer #2 (Recommendations for the authors):*

The manuscript introduces LabNet as a network-based platform for the control of hardware in Neuroscience. The authors recognize and attempt to address two fundamental problems in constructing systems neuroscience experiments: on one hand the importance of precise timing in the measurement and control of behavior; on the other hand, the need for flexibility in experimental design. These two goals are often at great odds with each other. Precise timing is more easily achieved when using fewer, dedicated homogeneous devices, such as embedded microcontrollers. Conversely, flexibility is more easily found in the diversity of devices and programming languages available in personal computers, but this often comes at the cost of a non-real-time operating system, where timing can be much harder to predict accurately. There is also a limitation on the number of devices which can be simultaneously controlled by a single processor, which can be an impediment for high-throughput behavior studies where the ability to run dozens of experiments in parallel is desirable.

LabNet proposes to address this tension by focusing on the design of a hardware control and instrumentation layer for embedded systems implemented on top of the Raspberry π family of microprocessors. The idea is to keep coordination of experimental hardware in a central computer, but keep time-critical components at the edge, each node running the same control software in a Raspberry π to provide precise timing guarantees. Flexibility is provided by the ability to connect an arbitrary number of nodes to the central computer using a unified message passing protocol by which the computer can receive events and send commands to each node.

The authors propose the use of the C++ programming language and the actor-model as a unifying framework for implementing individual nodes and present a series of benchmarks for evaluating the performance of LabNet in the Raspberry Pi, followed by a series of benchmarks comparing LabNet against other established hardware control platforms.

The first set of benchmarks is presented in Figure 2 and is used to understand how LabNet retains its performance across a variety of different Raspberry π hardware, and clients written in different languages (C++, C# and Python). Different tests are used to measure latency over the network (Set digital out test), and full round-trip latency by using a digital input event to directly trigger a digital output through LabNet (Read and set GPIO test).

A new stress test is also introduced in this revised manuscript to evaluate how many pins in parallel can be monitored by LabNet. Unfortunately, the plot in panel C is confusing, since the text mentions a measure in events per second, but the plot seems to have the same range of values as the other panels in the figure, and the axes are not labelled, so it is not clear whether we are watching a drop in processed events per second, or a drop in latency. This should be clarified, and all axes labelled accordingly.

The methodology of the tests is hard to follow from the text description alone. It would be useful to have a schematic diagram of the wiring used in the test, as well as an example interaction diagram with a timeline of how the different events in each component (PC and Raspberry Pis) trigger each other and which time intervals are being measured.

A second set of benchmarks is then presented in Figure 3 comparing LabNet to other established hardware control platforms. These are used to inform how well LabNet running over the network compares to different platforms running on local hardware.

In the revised manuscript, the authors use the same "read and set GPIO" test used for Figure 2. The authors demonstrate that LabNet achieves similar latencies to the local platforms despite hardware control running over the network. The tests are fair, and the results support the authors' conclusion, although it would have been preferable to include an independent measure of electrical signal timing for the Read and set GPIO test through an oscilloscope to exclude possible confounds with different software timestamping strategies of events in the different platforms tested (see for example Akam et al., 2022).

The manuscript concludes with a Discussion section highlighting the possibilities for interfacing with different hardware using existing LabNet adaptors for the Raspberry Pi. Here I remain at odds with the authors as I don't believe they have convincingly demonstrated their stated aim of ease of extensibility either in the client or in the server, which I feel is crucial to introduce LabNet as a platform for the open-source community.

Some examples of operant boxes for rodent experiments controlled by LabNet are presented in Figure 1. A variety of different devices are represented in the example schematics, mostly digital inputs and outputs, but also a visual display monitor and a tone generator, both mentioned in the discussion, the latter also in the code listings. The variety and heterogeneity of hardware components in such tasks is typically challenging to coordinate and synchronize in a full experiment so it would be great to understand how exactly the authors envision LabNet to play a part in the assembly of such experiments.

Unfortunately, none of these complete examples are developed in the body of the text, and it is unclear whether these are actual experiments collecting data in the lab. The authors have argued that listings 1-3 present code examples of a simple "experiment". However, these seem to be mostly restricted to setting up and configuring individual modules and single commands in LabNet. It is not clear how these basic building blocks are expected to be used to coordinate multiple devices in the context of an application running a full experimental protocol, and what are the caveats and limitations in such a case from a developer's perspective.

Specifically, it would have been important to see how well the code listings 1-3 would scale up or integrate the control of a full behaviour task with multiple conditions, or how to synchronize and benchmark the system when integrating with other hardware, such as cameras or physiology recordings. Even if LabNet is just a small component of the final system, it would be important to describe more fully a few examples as this would allow the non-technical reader to make a quick assessment of the versatility of LabNet for different types of experiments and to make it clear where exactly LabNet fits in the design of an experimental rig.

Is the system targeting only rodent tasks or is it reasonable to adapt the system to work for *Drosophila* or zebrafish? Can nodes easily support the simultaneous generation of pulse trains, visual stimulation, and video capture, and if yes how many tasks should each node be responsible for to optimize control bandwidth? Given the generality of the Raspberry π platform, I feel it would be important for readers to find some of these answers in the manuscript, even if just in the form of suggestive examples, to clarify how LabNet might be positioned in the space of open-source hardware in neuroscience, now and in future versions. A table listing exactly which modules are available would also help to make users aware of the possibilities of the platform.

Alternatively, given the experience of the authors in developing embedded animal behaviour platforms for several years, taking a few samples from that pool of experiments redesigned in LabNet would be very valuable to evaluate how this new software can help to address and alleviate common implementation bottlenecks.

Finally, the ease of the framework SObjectizer in extending LabNet is discussed. The reorganization of the document to move the implementation details into the Materials and methods section has overall made everything easier to read and follow.

I would have preferred, however, to see more examples on how to extend LabNet for a new custom device, compared to existing platforms such as Arduino or pyControl, in addition to the lengthier discussion of actor models. Even though a software plug-in system is not currently supported, it is possible to extend LabNet by modifying its source code. However, all listings in the paper currently target only the PC client side of LabNet. It would be great to see a few examples of the server side, or a schematic discussion on how to integrate new hardware. This would give a more practically grounded introduction to the actor model and help the reader more easily understand how to leverage and modify the LabNet open-source code for their specific purposes.

I fully agree that a robust, high-performance, and flexible hardware layer for combining neuroscience instruments is desperately needed, and so I do expect that a more thorough treatment of the methods developed in LabNet will in the future have a very positive impact in the field.

---

## [Author Response]

[Editors’ note: what follows is the authors’ response to the second round of review.]

– Remove language about other packages not meeting their claims re: stress test (as they were not tested).

I have removed "unlike others" in line 17. This did not refer to performance, but to the number of IOs that can be controlled simultaneously. For example, Bpod natively has only 2 inputs and 2 outputs, more only via module ports. Also, monitoring and simultaneous reaction to multiple events is possible, but relatively complicated (not only in Bpod). This has to do with the state machine. That is why other packages were not subjected to the stress test: number of IOs is not comparable. Nevertheless, my statement was misleading and has been adjusted.

– Clarify wording on stress test results (see R#2 review).

The description of the “stress” test is reworked. All y axes have a label.

– Describe means of measuring timings on both LabNet and other packages.

A diagram is added as an illustration. Tests description of the “read and set GPIO” is reworked.

– Link to specific versions of code for each of the tests.

Source code for Autopilot, Bpod and pyControl tests is inside the article’s data and source code repositoty. See the “key resources table” and “Code availability” sections. Previously inside the zip archive, now as clear text inside extra folders (autopilot, bpod and pyControl).

– Describe versions (with git hashes or semver) of all software, LabNet and other packages.

See “key resource table”.

– Respond to R#1 questions about example tests for other packages.

All tests for other tools are runnable. The measurement for all tools (including LabNet) was done in same way.

– Describe an example experiment done within the lab using LabNet and how it fits into the rest of the experimental setup.

A description of some so far built systems is now at the beginning of the discussion chapter.

– Label the axes in all figures.

Done

– Clarify the benchmarking protocol well, with a diagram.

Done

– Please include a key resource table.

“key resource table” is now included at the beginning of the “Materials and methods” section. It contains links to repositories of all tested packages with SHA hashes and versions if available.

Reviewer #1 (Recommendations for the authors):General comments:– Many of these will read as negative, so I want to start by saying I appreciate the author's work, apologize for the lateness of my review (life has been hectic pre- and post-dissertation!), and thank them for writing this package! All that I don't comment on here I think is good.– In general I like to see software timestamp measurements supplemented with hardware measurements (from eg. an oscilloscope), even just to confirm that the software timestamps are close. I don't think it's of huge importance here, but I wanted to make that future recommendation to the authors, especially when taking timestamps from an interpreted language like Python.

We can verify the results very easily. We know that the Bpod state machine is running at 10kHz. My test result for Bpod is 0.1ms with nearly 0 STD. Which is exactly the same as 10kHz. My test result for pyControl is very close to the results reported in the pyControl article in *eLife*. Additionally, I connected on the RasPi used for the measurements two pins together; this was to verify how fast the measurement software can detect input events. The result was 1 microsecond (also reported in the paper). And, of course, this measurement software was written in C++ (not Python) and ran on its own RasPi with no other software running on it. Thus, the RasPi functioned as a 1MHz oscilloscope and we can be sure that the measurements are correct. In addition, all software packages were tested in the same way. If there was a time offset, it was the same for pyControl, Bpod, Autopilot and LabNet.

– The mismatch between L84-L88 and the results is made more salient with the addition of L143-L147 – L84-88 say Python is intrinsically slow and thus C++ was chosen, but then L143-147 say that Python has an advantage because the C++ implementations are more complex. L84-88 thus read like theoretical concerns that were demonstrated here to not be true because of additional details in the C++ implementation, right?

The statement that Python is slower than C++ is of a more general nature. Python is normally executed by CPython. Critical and most important parts in CPython are implemented in C. This is also true for many important packages like NumPy. Thus, as long as a Python program does not require multithreading, has very few calculations directly in the Python code (outsource to C, e.g. via NumPy) and does not have a complex program flow, it can be just as performant as C++. Unfortunately, this is only true for simple programs, like our client implementation used for the article.

– I appreciated the expanded discussion of the intended uses for the package, like the discussion of the potential for using multiple pis together, etc. I think that and the brief descriptions of potential tasks help the paper!

The discussion now contains an overview and description of systems that have already been built using LabNet.

– I don't see a discussion of documentation in the main text, I don't think it's worth holding the paper up over, but I again make the recommendation to the authors to at least discuss their plans for documentation and future maintenance, as that is really the critical factor for whether a package like this will be adoptable by other labs. The authors briefly address this in their response, but yes this is important information for prospective users to have!

The documentation is now addressed at the end of the discussion. It is planned with much more detail for the next major version of LabNet, which is also already in development.

– Some of the other concerns that I raised in the prior recommendations for the authors were not addressed, perhaps that was my fault in not understanding how the public review. vs recommendations to authors work at eLife.

We have been careful to consider all points raised. With that in mind we have again carefully checked the manuscript.

Figure comments:Additional comments on new text:– Listings: The inline comments and in-text descriptions are much appreciated!– Figure 1: You designed that in TikZ? I am amazed. I would love to see that code. I checked out the TikZ code for the other figures and am very impressed.

source code for all figures is now included.

– Figure 2b-d: The y-axes are unlabeled.

now all y-axes have labels

– L17-18: I don't see stress test comparisons for the other packages, so the "unlike others" doesn't seem to be supported by the text.

removed, more in first response

– L18: typo, latenies -> latencies.– L61-63: This seems like an odd definition of openness to me, which typically means either that the source is inspectable. I would call the "control an experimental chamber on its own" part independence or modularity, and the "or together with a number of other nodes" interoperability or scalability. I am unsure how one would use multiple LabNet nodes in the same task, as an example doesn't seem to be in the text! This also seems to contradict L66-67 "However this comes with the restriction that at most one experimental system can be connected to each RasPi" – what counts as an experimental system here? are the authors just referring to a particular set of hardware components which could be combined in a single experimental chamber? that clarification would resolve the conflict to me.

The definition of openness and scalability refers to the distributed network and not to the source code. The definition also comes from Tanenbaum's book. Anyway, I adjusted the text here a little, hopefully it makes it clearer.

Examples of experimental systems are shown in Figure 1. Now also in Figure 4. The discussion contains now a brief description of so far built and with LabNet controlled system. But actually, anything can be an “experimental chamber”. Even Listings 1-3 already describe a system; a poke sensor, a valve and an LED. Small and simple experiments can already be realised with this hardware.

What is actually described here is that we want to bring any number of experimental systems into a network and then control them simultaneously. With one or multiple clients.

– L71-72: I am not sure what this means, the client is the controlling computer, but not sure what a task is in this context. And I thought that the hardware control happened on the raspi (server?).

Changed “task” to “duty”. Actually, the “task” or “duty” is in the first part of the sentence: “hardware control in the context of the experiments”. I added a new sentence.

– L72-73: from what I recall you also provide clients in these languages? might be worth some clarification describing what you mean by writing clients in multiple languages – eg. that clients written in multiple languages can interact with the underlying C++ library?

The sentence was moved to the end of the paragraph and now explicitly refers to Protobuf.

– L171: I found the description of the new crossover-based tests a bit hard to follow and it took me a few reads, I think a diagram would be helpful here :

The description is reworked. A diagram is added as an illustration.

– L210-221: I can't really tell how the code works for the new tests, it would be really helpful to link to the source code for each of the tests in the main text so we know what you're referring to! for example, the stress test looks like it does not send instructions of the network but operates on local logic as well: https://github.com/WinterLab-Berlin/LabNet/blob/3963f3371610d828e44af1e27ba6374cacc79748/examples/cpp/perf_test/main.cpp

The link leads to the measurement software that ran on the second RasPi. It generates the test signal and waits for the reaction from the LabNet, Autopilot, etc and saves the measured latencies locally in a csv file. It has no LabNet dependencies.

I see the data availability statement and found the repo, but it would be nice to have those separated out instead of inside of a zip file so you could link to them.

Source code for Autopilot, Bpod and pyControl tests is now inside extra folders in the repository (autopilot, bpod and pyControl).

– L228-229: The inclusion of whisker now reads as odd, since I think assuming latencies based on polling frequency is probably a pretty bad assumption in most cases.

Polling frequency normally gives the worst-case times. But of course, there may be other factors which can impact the latencies.

– L218-241: I am not sure how the latencies are measured for the other systems, as it looks like you were taking software timestamps for the LabNet tests, but I don't see timestamps being taken in any of the other comparison tests, so I assume that external timestamps were being taken? It also seems like some external trigger would be needed in some of these tests as well (see below)? It is also important to validate any software timestamps with external hardware timestamps, and they shouldn't be mixed without validation (eg. software timestamps could either exaggerate or underestimate latencies depending on where they are taken). Some additional clarification is needed here.

Latency measurements with Autopilot, Bpod and pyControl were done in exactly the same way as with LabNet. The second RasPi with the same measurement software was used for all tools. The description at the beginning of “Comparison” chapter is changed to reflect this. And yes, an external trigger is needed in all tests, this was provided by the second RasPi. The second RasPi is also the time reference. And because it runs completely independently from all tools, the measurements are comparable and just as good as with an oscilloscope. The measured latency data for Autopilot, Bpod and pyControl are also a part of the second repository.

From my reading it doesn't look like the pycontrol or autopilot tests would work, but I am out of the lab and don't have a π to run them on myself.

All tests are runnable. After the latency time tests, the source code was added to the repository without modifications. The pyControl test is very simple, see my short description below, but needs a bit of pyControl API knowledge. The Autopilot test is a simplified version of the “free water task”, from the official Autopilot repository.

For the pycontrol test, it seems like it would go- on_state(event='entry') -> p_off- off_state(event='p_off') -> goto on_state- on_state(event='entry') …and if the 'on' state was triggered manually from an external trigger it would go like- on_state(event='p_on') -> goto off_state- off_state(event='entry') -> switch p_out.on()- on_state(event='p_on') -> goto off_state- off_state(event='entry') …But I admit I am not altogether familiar with pycontrol. Some comments in the source would be lovely.

The pyControl test is very simple. “p_out” is a digital output, acts as response to the external event. “p_in” is digital input with “p_on” as rising and “p_off” falling events. “p_in is also the external trigger event which comes from the measurement RasPi. Short state machine description:

– enter “on_state” -> turn “p_out” off

– wait until external trigger “p_on” -> go to the state “off_state”

– enter “off_state” -> turn p_out on

– wait until external trigger “p_off” -> go to the state “on_state”

– repeat from beginning

For the Autopilot test, there are two stage methods in a cycle, water() and response(). the water method clears the stage_block which would cause the running pilot to wait for any set triggers. A trigger is set for the 'sig_in' pin that should set the 'sig_out' pin high. When the sig_in trigger is received, that should cause the response() method to be called which sets 'sig_out' low after some delay and then return to the water stage immediately. This would require some external input to trigger the sig_in event, and then the timestamps of the external input and the sig_out pin would be compared. If the sig_out pin were wired to sig_in, the test wouldn't work, as the sig_out pin would never be set high. Having the `prefs.json` file from the pilot would be useful to include here to avoid ambiguity in system configuration, as I am assuming the default polarity configuration settings are used on the digital input and output classes.

The file “prefs.json” is now included and inside the “autopilot” folder. As with other tools Autopilot’s RasPi needs to be connected with 2 pins to the measurement RasPi. Autopilot needs to wait for the external events and reacts to them. The time is only measured on the second RasPi, not on Autopilot’s RasPi.

A set of tests that are more similar to the tests described for labnet are available in the plugin accompanying our manuscript: https://github.com/auto-pi-lot/plugin-paper/blob/c6263a4890b7d6101688158d8acb3aaeb9199533/plugin_paper/scripts/test_gpio.py documented here: https://wiki.auto-pi-lot.com/index.php/Plugin:Autopilot_Paper We find the roundtrip (oscilloscope measurement of input to output) latency to be 400us.

These tests contains nearly no Autopilot functionality. E.g. in the "test_readwrite" test, a function is simply assigned to the pigpio callback and pigpio automatically calls this function when a signal is present. Thus, it tests only the speed of pigpio and its Python interface. I don't think real experiments in Autopilot work that way.

The “read and set GPIO” in our paper produces much more realistic latencies. Simply because all tools, including LabNet, have to use their regular logic like they would to run experiments.

I don't doubt the authors ran the tests, and please correct where my read of the code is wrong, but I think some additional detail is needed in the reporting of the results in any case.

I think that the test description in the previous version of the manuscript was misleading. It has now been rewritten. The former version arose because I wanted to describe the connection between the measuring RasPi and LabNet/Bpod/pyControl/Autopilot very precisely. Unfortunately, it became too complicated. Actually, there are only two pins connected. One as an external trigger and the second as a response. The tests as described now should be understandable.

Reviewer #2 (Recommendations for the authors):The manuscript introduces LabNet as a network-based platform for the control of hardware in Neuroscience. The authors recognize and attempt to address two fundamental problems in constructing systems neuroscience experiments: on one hand the importance of precise timing in the measurement and control of behavior; on the other hand, the need for flexibility in experimental design. These two goals are often at great odds with each other. Precise timing is more easily achieved when using fewer, dedicated homogeneous devices, such as embedded microcontrollers. Conversely, flexibility is more easily found in the diversity of devices and programming languages available in personal computers, but this often comes at the cost of a non-real-time operating system, where timing can be much harder to predict accurately. There is also a limitation on the number of devices which can be simultaneously controlled by a single processor, which can be an impediment for high-throughput behavior studies where the ability to run dozens of experiments in parallel is desirable.LabNet proposes to address this tension by focusing on the design of a hardware control and instrumentation layer for embedded systems implemented on top of the Raspberry π family of microprocessors. The idea is to keep coordination of experimental hardware in a central computer, but keep time-critical components at the edge, each node running the same control software in a Raspberry π to provide precise timing guarantees. Flexibility is provided by the ability to connect an arbitrary number of nodes to the central computer using a unified message passing protocol by which the computer can receive events and send commands to each node.The authors propose the use of the C++ programming language and the actor-model as a unifying framework for implementing individual nodes and present a series of benchmarks for evaluating the performance of LabNet in the Raspberry Pi, followed by a series of benchmarks comparing LabNet against other established hardware control platforms.The first set of benchmarks is presented in Figure 2 and is used to understand how LabNet retains its performance across a variety of different Raspberry π hardware, and clients written in different languages (C++, C# and Python). Different tests are used to measure latency over the network (Set digital out test), and full round-trip latency by using a digital input event to directly trigger a digital output through LabNet (Read and set GPIO test).A new stress test is also introduced in this revised manuscript to evaluate how many pins in parallel can be monitored by LabNet. Unfortunately, the plot in panel C is confusing, since the text mentions a measure in events per second, but the plot seems to have the same range of values as the other panels in the figure, and the axes are not labelled, so it is not clear whether we are watching a drop in processed events per second, or a drop in latency. This should be clarified, and all axes labelled accordingly.

The text has now been modified: the results in the figure are described briefly first, then the events per second.

All y axes now have a label.

The methodology of the tests is hard to follow from the text description alone. It would be useful to have a schematic diagram of the wiring used in the test, as well as an example interaction diagram with a timeline of how the different events in each component (PC and Raspberry Pis) trigger each other and which time intervals are being measured.

The test descriptions have been changed and simplified. They should now be easier to understand. A time signal diagram is now included.

A second set of benchmarks is then presented in Figure 3 comparing LabNet to other established hardware control platforms. These are used to inform how well LabNet running over the network compares to different platforms running on local hardware.In the revised manuscript, the authors use the same "read and set GPIO" test used for Figure 2. The authors demonstrate that LabNet achieves similar latencies to the local platforms despite hardware control running over the network. The tests are fair, and the results support the authors' conclusion, although it would have been preferable to include an independent measure of electrical signal timing for the Read and set GPIO test through an oscilloscope to exclude possible confounds with different software timestamping strategies of events in the different platforms tested (see for example Akam et al., 2022).

See my responses above to the first reviewer.

The manuscript concludes with a Discussion section highlighting the possibilities for interfacing with different hardware using existing LabNet adaptors for the Raspberry Pi. Here I remain at odds with the authors as I don't believe they have convincingly demonstrated their stated aim of ease of extensibility either in the client or in the server, which I feel is crucial to introduce LabNet as a platform for the open-source community.Some examples of operant boxes for rodent experiments controlled by LabNet are presented in Figure 1. A variety of different devices are represented in the example schematics, mostly digital inputs and outputs, but also a visual display monitor and a tone generator, both mentioned in the discussion, the latter also in the code listings. The variety and heterogeneity of hardware components in such tasks is typically challenging to coordinate and synchronize in a full experiment so it would be great to understand how exactly the authors envision LabNet to play a part in the assembly of such experiments.

The synchronisation is automatically given from the order in which the events (also from several RasPis) arrive. The coordination is given through the experiment logic.

Unfortunately, none of these complete examples are developed in the body of the text, and it is unclear whether these are actual experiments collecting data in the lab. The authors have argued that listings 1-3 present code examples of a simple "experiment". However, these seem to be mostly restricted to setting up and configuring individual modules and single commands in LabNet. It is not clear how these basic building blocks are expected to be used to coordinate multiple devices in the context of an application running a full experimental protocol, and what are the caveats and limitations in such a case from a developer's perspective.

Unfortunately, it would be beyond the scope of this article to describe the client side of the experiments. The focus of the current article is clearly the core LabNet functionality that is independent of specific client implementations. With listings 1-3 we describe exactly what a developer of client software has to do: open TCP/IP connections to all LabNet/RasPi devices, initialize hardware, communicate with hardware. It does not matter how much hardware is connected on how many RasPis the logic is always the same.

Specifically, it would have been important to see how well the code listings 1-3 would scale up or integrate the control of a full behaviour task with multiple conditions, or how to synchronize and benchmark the system when integrating with other hardware, such as cameras or physiology recordings. Even if LabNet is just a small component of the final system, it would be important to describe more fully a few examples as this would allow the non-technical reader to make a quick assessment of the versatility of LabNet for different types of experiments and to make it clear where exactly LabNet fits in the design of an experimental rig.

A description of some so far built systems is now at the beginning of the discussion chapter.

Is the system targeting only rodent tasks or is it reasonable to adapt the system to work for *Drosophila* or zebrafish? Can nodes easily support the simultaneous generation of pulse trains, visual stimulation, and video capture, and if yes how many tasks should each node be responsible for to optimize control bandwidth? Given the generality of the Raspberry π platform, I feel it would be important for readers to find some of these answers in the manuscript, even if just in the form of suggestive examples, to clarify how LabNet might be positioned in the space of open-source hardware in neuroscience, now and in future versions. A table listing exactly which modules are available would also help to make users aware of the possibilities of the platform.

We think that the large advantage of LabNet is that it can be used universally for laboratory experimental automation, both within and beyond neuroscience. It is agnostic to the species under study and we have used it from bats to flies. A description of some so far built system is now at the beginning of the discussion chapter.

The LabNet architecture with actors targets explicitly the execution of multiple tasks. The Discussion chapter contains now a small discussion about performance and bandwidth. So far, we never had performance or bandwidth issues with LabNet/RasPi. But this excludes applications with video acquisition. A list with already available interfaces is in the “Implementation” subchapter.

Alternatively, given the experience of the authors in developing embedded animal behaviour platforms for several years, taking a few samples from that pool of experiments redesigned in LabNet would be very valuable to evaluate how this new software can help to address and alleviate common implementation bottlenecks.

We wanted to address two points with LabNet: performance and the possibility to control a large number of experimental systems. We succeeded (as we think) in both and both points are now discussed in more detail in the paper.

Finally, the ease of the framework SObjectizer in extending LabNet is discussed. The reorganization of the document to move the implementation details into the Materials and methods section has overall made everything easier to read and follow.I would have preferred, however, to see more examples on how to extend LabNet for a new custom device, compared to existing platforms such as Arduino or pyControl, in addition to the lengthier discussion of actor models. Even though a software plug-in system is not currently supported, it is possible to extend LabNet by modifying its source code. However, all listings in the paper currently target only the PC client side of LabNet. It would be great to see a few examples of the server side, or a schematic discussion on how to integrate new hardware. This would give a more practically grounded introduction to the actor model and help the reader more easily understand how to leverage and modify the LabNet open-source code for their specific purposes.

Unfortunately, this would have become very technical. Readers would need to know C++ and treating the topic comprehensively would need about 5-10 more pages. In the case of actors, this would also just be a repetition of the many books written about this. In the case of the extensibility of LabNet, I would only describe the current status, which may already be different in the next version. In any case, that will be the case as soon as the plug-in system has been implemented. We fully agree that this information should be available. However, our decision has been to place this information in the online documentation. This is the main reason that we have remained more general in the methods section.